# The Limited Responses of Provincial Expressway Network Operation Quality to the Impact of COVID-19: Taking Shaanxi Province as an Example

**Yongji Ma** [1,*], **Jinliang Xu** [1,*], **Chao Gao** [1], **Chenwei Gu** [1] and **Xiaohui Tong** [2]

1   School of Highway, Chang'an University, Xi'an 710064, China
2   China Communications Construction First Highway Consultants Co., Ltd., Xi'an 710061, China
*   Correspondence: 2021021055@chd.edu.cn (Y.M.); xujinliang@chd.edu.cn (J.X.)

**Abstract:** Recent research indicates that the coronavirus disease 2019 (COVID-19) pandemic has impacted road traffic operation quality. From the perspective of service level, safety level, and operation orderliness, this study attempts to analyze the limited responses of the operation quality of the provincial expressway network (PEN) to the impact of COVID-19. The traffic operation data of the PEN in Shaanxi Province were collected for three consecutive years (from 2019 to 2021) before and after the COVID-19 outbreak, including weekly average daily traffic, weekly traffic interruption times, weekly traffic control time, weekly traffic accidents, weekly traffic injuries, and weekly traffic deaths. Through descriptive statistics and dynamic analysis, we studied the limited responses of the safety and service levels of the PEN to the impact of COVID-19. An evaluation model of operation orderliness of the PEN was established by using dissipative structure theory and entropy theory. Results show that in 2020, the service level, safety level, and operation orderliness of the PEN dropped to their lowest. With the gradual control of pandemic, its negative impacts on the service level and operation orderliness of the PEN gradually decreased but did not recover to the pre-pandemic level. In terms of the safety level, the number of traffic accidents, traffic injuries, and traffic deaths decreased, whereas the traffic accident rate and accident casualty rate remained high. In summary, the operation quality of the PEN in Shaanxi Province has had a limited response to the impact of COVID-19.

**Keywords:** provincial expressway network; operation quality; service level; safety level; operation orderliness; COVID-19 pandemic; limited responses

## 1. Introduction

The worldwide COVID-19 outbreak has brought great challenges to the social development of countries globally [1–6]. In China, there have been more than five million confirmed cases and more than 23,000 deaths since the start of the COVID-19 outbreak [7]. Facing the threat of pandemic, many countries, including China, took effective pandemic prevention and control measures to protect the health and life safety of residents [8–13]. Among them, several orders and calls, such as stay-at-home orders, the call for reducing gathering and "only go out for essential trips" reduced people's travel frequency to a certain extent. In this context, the characteristics of expressway traffic operation were bound to change, which would further affect the road traffic quality.

Many researchers studied the impacts of COVID-19 on road traffic from different perspectives. Yasin et al. reviewed the effects of the COVID-19 pandemic on the incidence, patterns, and severity of the road traffic collisions. They found that the COVID-19 pandemic reduced the overall absolute numbers of road traffic collisions despite the relative increase in severity of injury and death [14]. Zhang et al. investigated the impact of COVID-19 on the number of people involved in crashes, accounting for the intensity of different pandemic control measures. They identified the negative relationship between stay-at-home policies implemented and the number of people involved in crashes in New York

City [15]. Sekadakis et al. studied the impact of the COVID-19 pandemic on road traffic collisions, fatalities, and injuries through time series analyses. They found that the road traffic volume decreased sharply during the lockdown period, whereas the fatalities and slightly injured rates significantly increased [16].

In addition, Paramasivan et al. examined the impact of the pandemic-induced lockdown on road traffic deaths and injuries in Tamil Nadu, India. They found a substantial reduction in traffic crashes, injuries, and fatalities during lockdowns, whereas the number of grievous injuries reduced more than minor injuries, crashes, or fatalities [17]. Zhao et al. analyzed the time distribution characteristics of traffic accidents under the pandemic situation through traffic accident alarm data. They found that the traffic control policies implemented in the early stage of the pandemic reduced the traffic accident volume to some extent [18]. Doucette et al. examined the impact of COVID-19's stay-at-home order on daily vehicle miles travelled (VMT) and motor vehicle crashes in Connecticut. The results indicated that the mean daily VMT significantly decreased in the post stay-at-home period in 2020, whereas several types of crash rates increased [19]. Vanlaar et al. compared self-reported risky driving behaviors (speeding, distracted driving, drinking and driving, and drugged driving) during the pandemic in Canada and the U.S. They identified that U.S. drivers were more likely to engage in risky driving behaviors during the pandemic than those in Canada [20]. Qureshi et al. performed an in-depth analysis using data from the Statewide Traffic Accident Records System in Missouri. They found that the mandated societal lockdown policies led to a reduction in road traffic accidents, resulting in non-serious or no injuries but not those resulting in serious or fatal injuries [21]. In general, these above-mentioned studies came from different countries and regions, all of which proved that the COVID-19 pandemic and its traffic control policies had a non-negligible impact on road traffic safety.

Moreover, Tucker et al. studied the proportion of motor vehicles speeding. They found that both the percentage of cars engaged in extreme speeding and those traveling below the speed limit increased during the COVID-19 stay-at-home period [22]. Patra et al. investigated the short-term changes in road traffic patterns in the COVID-19 pandemic context. They demonstrated that road traffic activities significantly reduced due to the restrictions in non-essential trips, workplace suspensions, and strict surveillance during lockdowns [23]. Muley et al. assessed the impact of pandemic response measures on the road traffic mobility by quantifying the holistic impact of the incremental measures at different stages on traffic volumes and safety [24]. Ghanim et al. established a model to evaluate the impact of COVID-19 pandemic prevention and control policies on the urban traffic flow [25]. Macioszek et al. analyzed the road traffic volume in Gliwice (Poland) before and during the restrictions related to COVID-19 [26]. Their findings indicated that COVID-19 reduced the road traffic volumes in Poland, which was similar to many other countries, and the different shape and duration of afternoon peaks and morning peaks was the same for 2019 and 2020 but radical changes were noted during the pandemic. Parr et al. studied the statewide highway volume data in Florida in the COVID-19 pandemic context. They found that the overall statewide traffic volume dropped and there were differences between highways and arterials both in terms of the timing and extent [27]. Liu et al. demonstrated a reduction in road traffic volume in Minnesota in the COVID-19 context [28]. Simunek et al. analyzed the main impacts of COVID-19 movement restrictions on the road traffic in the Czech Republic. They proved the impacts of traffic restriction policy on road traffic, including the reduction of traffic density and the increase in traffic speed [29].

Besides, Dong et al. investigated how the COVID-19 pandemic impacted the likelihood of severe crashes via changing driving behaviors. The results suggested that aggressiveness and inattentiveness of drivers increased significantly after the outbreak of COVID-19, leading to a higher likelihood of severe crashes [30]. Katrakazas et al. compared observed values concerning three traffic indicators (average speed, speeding, and harsh braking), with forecasts based on their corresponding observations before the lockdown in Greece. Results indicated that speeds and harsh braking were found to increase compared to

the forecasted evolution, while road crashes in Greece were reduced [31]. Nomura et al. estimated the exiguous deaths from road injuries in each week from January to September 2020 by 47 prefectures. They found that the number of road traffic fatalities during the COVID-19 pandemic in Japan decreased slightly, but not significantly [32]. Truelove et al. explored how speeding and phone use while driving (road behaviors associated with a high crash risk) changed as a result of COVID-19 restrictions in Queensland. Their findings indicated that self-reported speeding and phone use significantly decreased during restrictions, but not for those who continued to drive regularly [33].

Taken together, the impacts of the COVID-19 pandemic and its prevention and control measures on road traffic are non-negligible. From another aspect, existing research has mainly focused on the theme of traffic safety and traffic volume. A few studies involved traffic speed, traffic density, and microscopic traffic behaviors. Research on the traffic quality of provincial road networks from the perspective of multiple indicators and multiple parameters is limited, which is precisely the main problem to be solved in this study.

Furthermore, road traffic exists as a very complex system, and many methods and theories exist to study it. Among them, theories to study the development state of material systems, entropy theory and dissipative structure theory have been applied by researchers in the field of traffic engineering. For example, Chai et al. used information entropy theory to build a composite element evaluation model for urban mountain road safety [34]. Sun et al. set up an urban traffic congestion prediction model through the hidden Markov model combined with dissipative structure theory and entropy theory [35]. Deng et al. applied the information entropy theory to the study of road traffic safety evaluation in Jiangsu Province and obtained reasonable evaluation results [36]. Hence, entropy theory and dissipative structure theory can be used to study the road traffic system, and we finally chose them as part of the research methods in this study.

Taking the service level, safety level, and operation orderliness as the points of penetration, this study attempts to analyze the traffic operation quality of the PEN in Shaanxi Province in the COVID-19 pandemic context. Through statistical analysis and modeling research, the limited responses of the PEN operation quality to the impact of the COVID-19 pandemic were explored. By quantitative analysis, the negative impacts of the pandemic on expressway traffic were identified, which could provide relevant theoretical references for the decision-making of traffic management departments in improving road operation quality.

## 2. Materials and Methods

In 2021, there were 19 expressways in the PEN in Shaanxi Province, with a total mileage of 6484 km. Figure 1 depicts its geographical location overview.

We collected the operation data of the PEN for three consecutive years (from 2019 to 2021) before and after the COVID-19 outbreak, including weekly average daily traffic (WADT), weekly traffic interruption times (WTIT), weekly traffic control time (WTCT), weekly traffic accidents (WTA), weekly traffic injuries (WTI), and weekly traffic deaths (WTD). Among them, WTIT refers to the number of traffic interruptions caused by traffic accidents in the PEN in a week. The higher the value, the more frequent the PEN interruptions in that week, and vice versa. WTCT indicates the traffic control time caused by traffic accidents in the PEN in a certain week. When a road accident occurs, to ensure accident rescue and orderly traffic, the traffic management departments will temporarily take measures to control the accident road section, restrict the passage of a certain vehicle type, take one-way alternate passage measures, and others. The total duration of traffic control in a week is WTCT.

The operation orderliness of the PEN examined in this study refers to the degree to which the traffic flow can operate continuously, stably, and smoothly under certain traffic conditions. If the traffic flow can operate continuously, stably, and smoothly, the operation orderliness of the PEN is high, and the traffic condition is good. Otherwise, the operation orderliness is low, and the traffic condition is poor.

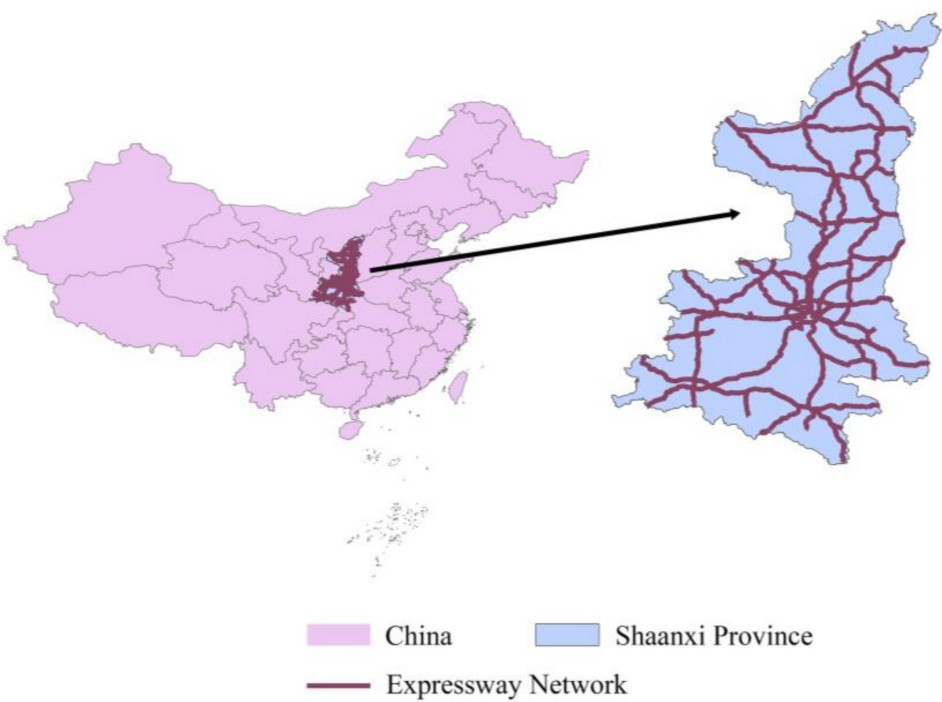

**Figure 1.** The geographical location overview of the PEN in Shaanxi Province.

The PEN operation quality in the COVID-19 pandemic context was analyzed from the perspective of service level, safety level, and operation orderliness.

Taking WADT, WTIT, and WTCT as the PEN service level evaluation parameters, and WTA, WTI, and WTD as the PEN safety level evaluation parameters, the limited responses of the PEN service and safety levels to the impact of the COVID-19 pandemic were studied through dynamic analysis and descriptive statistics.

Taking WADT as the core parameter and WTIT as the system entropy weight parameter, an evaluation model for operation orderliness of the PEN was established. On this basis, the entropy changes in the PEN system before and after the COVID-19 outbreak were analyzed, and the limited responses of the PEN operation orderliness to the impact of the COVID-19 pandemic were studied.

*2.1. Materials*

We collected the traffic data from 127 continuous observation sites of 19 expressways in Shaanxi Province, including WADT, WTIT, WTCT, WTA, WTI, and WTD, from 2019 to 2021. These 127 continuous observation sites were mainly distributed at the interchanges, bridges, and long tunnel sections. The automatic traffic flow observation instruments were mainly used to count the traffic volume, and the data obtained were finally published on the official website of the Department of Transport of Shaanxi Province. Other traffic data analyzed in this study were counted by local traffic management departments and finally reported to the provincial traffic department for a summary. These data were released together with traffic volume data. All traffic interruptions and traffic controls were caused by traffic accidents rather than other emergencies. These traffic statistics covered the whole PEN and could represent the overall level of the provincial expressway traffic operation within a certain period of time.

Affected by the pandemic, we did not obtain the complete data for 52 weeks of the whole year. From 2019 to 2021, statistics of 45, 34, and 39 weeks were collected, respectively. We numbered the data of each week in each year in chronological order and finally screened out the data of 78 weeks (26 weeks each year) with overlapping time points. We obtained the time series analysis data set of the PEN, as shown in Tables 1 and 2. This part of the data overlaps at time points, and the number of samples within the year is consistent,

which can provide more reliable data support for interpreting the PEN traffic operation in the COVID-19 pandemic context. Meanwhile, according to the records of the traffic management departments of Shaanxi Province, the weather conditions in the period to which this part of the data belonged were the same, mainly sunny and cloudy days. Besides, there was no influence of major holidays, so the negative impacts of adverse weather and major holidays on the PEN operation quality were excluded, which could ensure the typicality and credibility of the traffic data.

**Table 1.** Time series analysis data set of the PEN service level evaluation parameters.

| Weekly Serial Number | WADT (pcu) | | | WTIT | | | WTCT (h) | | |
|---|---|---|---|---|---|---|---|---|---|
| | 2019 | 2020 | 2021 | 2019 | 2020 | 2021 | 2019 | 2020 | 2021 |
| 1 | 21,226 | 22,329 | 22,350 | 49 | 28 | 32 | 166.72 | 55.35 | 39.00 |
| 2 | 22,173 | 22,468 | 21,508 | 22 | 32 | 46 | 40.35 | 50.25 | 79.63 |
| 3 | 23,525 | 36,606 | 22,767 | 16 | 35 | 19 | 29.38 | 53.70 | 19.00 |
| 4 | 24,555 | 37,001 | 23,831 | 13 | 31 | 22 | 31.35 | 65.40 | 33.72 |
| 5 | 24,332 | 39,468 | 23,562 | 13 | 60 | 18 | 30.93 | 97.27 | 34.00 |
| 6 | 21,341 | 31,968 | 29,585 | 19 | 49 | 26 | 27.20 | 79.28 | 18.27 |
| 7 | 24,805 | 27,479 | 24,950 | 20 | 25 | 22 | 35.48 | 85.80 | 46.78 |
| 8 | 23,851 | 26,991 | 25,127 | 35 | 41 | 64 | 48.50 | 59.62 | 116.20 |
| 9 | 25,154 | 26,750 | 23,899 | 10 | 17 | 26 | 15.58 | 28.95 | 27.08 |
| 10 | 25,455 | 25,614 | 25,365 | 33 | 21 | 26 | 57.73 | 25.37 | 37.03 |
| 11 | 25,637 | 25,511 | 27,606 | 41 | 23 | 31 | 72.47 | 39.62 | 77.00 |
| 12 | 26,109 | 25,721 | 30,794 | 30 | 33 | 15 | 49.97 | 72.20 | 17.00 |
| 13 | 25,927 | 26,582 | 34,458 | 34 | 26 | 25 | 50.07 | 41.28 | 49.50 |
| 14 | 24,701 | 26,687 | 27,969 | 17 | 25 | 26 | 61.13 | 72.63 | 44.45 |
| 15 | 23,082 | 28,291 | 25,801 | 22 | 41 | 37 | 41.60 | 33.13 | 108.73 |
| 16 | 27,741 | 29,325 | 27,393 | 23 | 26 | 38 | 24.72 | 43.17 | 112.85 |
| 17 | 28,562 | 28,616 | 21,602 | 11 | 21 | 32 | 18.88 | 30.48 | 65.00 |
| 18 | 28,308 | 28,049 | 21,985 | 39 | 18 | 28 | 50.03 | 31.93 | 37.80 |
| 19 | 28,324 | 30,458 | 19,890 | 24 | 20 | 22 | 28.17 | 23.37 | 15.05 |
| 20 | 28,065 | 28,689 | 18,084 | 31 | 37 | 19 | 55.63 | 54.32 | 31.73 |
| 21 | 27,776 | 28,162 | 21,429 | 30 | 16 | 33 | 22.70 | 24.08 | 79.00 |
| 22 | 26,934 | 26,029 | 21,919 | 29 | 35 | 15 | 47.20 | 65.25 | 25.67 |
| 23 | 26,267 | 25,897 | 20,971 | 17 | 32 | 40 | 28.68 | 70.58 | 50.97 |
| 24 | 26,385 | 24,402 | 20,273 | 40 | 28 | 25 | 88.53 | 48.97 | 74.97 |
| 25 | 26,231 | 26,287 | 20,499 | 30 | 58 | 28 | 90.78 | 37.10 | 43.10 |
| 26 | 24,769 | 26,964 | 18,398 | 26 | 51 | 21 | 47.78 | 26.92 | 54.88 |

In China, faced with the threat of the pandemic, all provinces took effective measures to strictly prevent and control COVID-19. In the first half of 2020, Shaanxi Province activated the first- and third-level public health emergency responses successively. During this time, different traffic control measures were taken on the PEN in the whole province, which greatly impacted the expressway traffic characteristics. For example, all vehicles, except anti-pandemic-related vehicles, on expressways in the province were prohibited from passing during the implementation of the first-level public health emergency response. During the implementation of the third-level public health emergency response, people's travel restrictions were eased, but they were still not encouraged. Vehicle travel in medium- and high-risk regions was severely restricted during the subsequent regular pandemic prevention and control period. As a result, traffic in some areas was blocked, and the COVID-19 pandemic impacted the PEN operation quality to a certain extent under such circumstances.

**Table 2.** Time series analysis data set of the PEN safety level evaluation parameters.

| Weekly Serial Number | WTA | | | WTI | | | WTD | | |
|---|---|---|---|---|---|---|---|---|---|
| | 2019 | 2020 | 2021 | 2019 | 2020 | 2021 | 2019 | 2020 | 2021 |
| 1 | 147 | 101 | 105 | 33 | 35 | 25 | 3 | 5 | 4 |
| 2 | 60 | 80 | 165 | 22 | 43 | 38 | 4 | 6 | 1 |
| 3 | 56 | 109 | 77 | 21 | 11 | 13 | 4 | 1 | 5 |
| 4 | 44 | 105 | 95 | 11 | 33 | 22 | 1 | 4 | 6 |
| 5 | 74 | 147 | 80 | 8 | 28 | 32 | 3 | 8 | 1 |
| 6 | 68 | 133 | 110 | 27 | 37 | 36 | 0 | 6 | 3 |
| 7 | 93 | 66 | 86 | 37 | 24 | 28 | 3 | 2 | 0 |
| 8 | 88 | 82 | 135 | 33 | 25 | 31 | 4 | 4 | 3 |
| 9 | 66 | 47 | 99 | 23 | 15 | 46 | 1 | 2 | 1 |
| 10 | 87 | 91 | 104 | 25 | 22 | 37 | 2 | 5 | 3 |
| 11 | 108 | 80 | 102 | 31 | 22 | 39 | 5 | 0 | 5 |
| 12 | 76 | 89 | 74 | 28 | 32 | 18 | 2 | 4 | 4 |
| 13 | 89 | 64 | 86 | 27 | 17 | 19 | 3 | 2 | 6 |
| 14 | 50 | 97 | 85 | 32 | 31 | 27 | 8 | 5 | 5 |
| 15 | 75 | 87 | 94 | 25 | 32 | 32 | 3 | 1 | 8 |
| 16 | 75 | 91 | 73 | 23 | 34 | 20 | 3 | 6 | 10 |
| 17 | 27 | 88 | 139 | 9 | 25 | 26 | 1 | 2 | 1 |
| 18 | 76 | 63 | 79 | 24 | 25 | 17 | 3 | 2 | 0 |
| 19 | 63 | 78 | 54 | 25 | 27 | 10 | 5 | 7 | 1 |
| 20 | 88 | 93 | 78 | 18 | 28 | 19 | 4 | 2 | 2 |
| 21 | 69 | 78 | 109 | 25 | 32 | 24 | 1 | 1 | 7 |
| 22 | 78 | 99 | 77 | 36 | 13 | 14 | 3 | 3 | 2 |
| 23 | 77 | 118 | 75 | 23 | 28 | 20 | 3 | 11 | 3 |
| 24 | 86 | 157 | 76 | 25 | 28 | 25 | 3 | 1 | 4 |
| 25 | 80 | 124 | 79 | 31 | 21 | 18 | 3 | 2 | 6 |
| 26 | 50 | 140 | 52 | 8 | 23 | 13 | 2 | 6 | 2 |

*2.2. Methods*

The PEN is a dynamically evolving open system that covers drivers, vehicles, roads, environment, traffic management measures, and other elements. In the PEN operation process, vehicle driving is affected by road geometrics, weather conditions, traffic managements, and other factors. Drivers constantly obtain information from the external environment to control the operating state of vehicles. Car-following, lane-changing, overtaking, and other vehicle behaviors, have great randomness, leading to mutual interferences and dynamic changes of traffic flow. Based on these traffic characteristics of the PEN, we used dissipative structure theory and entropy theory to explore the limited responses of the PEN operation orderliness to the impact of the COVID-19 pandemic.

2.2.1. Entropy Theory and Dissipative Structure Theory

As a state function, entropy can represent the collective properties of abundant microscopic factors constituting the system from the macro level and reflect the disorder degree of the system. The higher the system entropy, the higher the disorder degree [37]. For an open system, the entropy change dS usually consists of two parts:

1. The entropy increase ($d_iS$) caused by irreversible processes inside the system.
2. Entropy change ($d_eS$) generated when the system exchanges matter and energy with the external environment.

The total entropy change of the system is the sum of the entropy changes of two parts [38]:

$$dS = d_iS + d_eS \tag{1}$$

In physics, a system can form a dissipative structure if it simultaneously satisfies the following four conditions [39]:

1. The system is open.

2. The system is far away from the equilibrium state.
3. Nonlinear interactions exist in the system.
4. The system has fluctuation phenomena.

The PEN has four basic conditions to become a dissipative structure, so the relevant theories are applicable. Based on the dissipative structure theory, the change of system orderliness can be measured by the system entropy change [40]:

1. When dS > 0, the total entropy of the system increases, and the system moves away from the equilibrium state and develops in the direction of disorder.
2. When dS < 0, the total entropy of the system decreases, and the system approaches the equilibrium state and develops in an orderly direction.
3. When dS = 0, the total entropy of the system remains unchanged, and the system can maintain a dynamic equilibrium state.

The calculation methods of system entropy mainly include thermodynamic-like method and statistical-physics-like method [40]. The former calculates the entropy value through the ratio of certain extensive and intensity quantities in the system, and the latter uses the information theory to calculate the entropy value. Considering that the PEN is not only a dissipative structure but also an information system, we finally calculated the PEN system entropy through information theory [41]. According to Shannon's information theory, if the output of an information source is finite or listable infinite discrete random variables, then it is discrete. The PEN can be regarded as a discrete information source. For a discrete information source X, let $x_i$ be the possible output signal of X ($x_i \in X$, $i = 1, 2, 3, \ldots, n$), and its probability is $p(x_i)$. Then, the calculation formula of system entropy value H(x) is as follows [42]:

$$\mathrm{H(x)} = -C \sum_{i=1}^{n} p(x_i) \log_{10} p(x_i) = \sum_{i=1}^{n} p(x_i) I(x_i), \tag{2}$$

where $C$ is a constant, $I(x_i)$ is the self-information of $x_i$, and the calculation formula of $I(x_i)$ is as follows:

$$I(x_i) = -C \log_{10} p(x_i). \tag{3}$$

The unit of $I(x_i)$ is Hartley.

### 2.2.2. Evaluation Model of the PEN Operation Orderliness

According to the dissipative structure theory, the first task of orderliness evaluation is to determine the order parameter in the model, because it determines the changes of other variables and affects the overall evolution process of the system. Its value determines the orderliness degree of the system, which can be used to characterize the orderliness degree and balanced development direction of the system [41]. Traffic volume is the most important characteristic parameter to the PEN system. Its amount and change characteristics directly determine the operation orderliness of the expressway, affect other traffic parameters, such as traffic speed and density, and reflect the developments of the regional economy and social environment to a certain extent. In this case, the order parameter involved should reflect the traffic volume characteristics when evaluating the PEN operation orderliness.

With regard to the PEN, its operation orderliness is related to various factors, such as traffic volume, traffic interruptions, traffic accidents, weather conditions, and others. In addition, the mileage of expressways and the ownership of private cars in Shaanxi Province have been growing rapidly under the influence of social development, which has non-negligible impacts on the traffic volume of the PEN.

Hence, the absolute value of traffic volume is not enough to reflect the operation orderliness level. To evaluate the operation orderliness more accurately, we introduced the patency degree λ as the order parameter of the PEN. Λ is defined as the ratio of daily average traffic volume within a certain period of a year (e.g., year m, m = . . . , 2019, 2020,

2021, . . . ) to the annual average daily traffic of the same year. For the PEN at week $i$ of a year, the patency degree $\lambda$ can be defined as:

$$\lambda_i = \frac{W_i ADT}{AADT}, \tag{4}$$

where $i$ represents the weekly sequence number, and $W_i ADT$ represents the WADT of week $i$. If $\lambda_i = 1$, the provincial traffic volume level in week $i$ is consistent with the annual average level. If $\lambda_i < 1$, the provincial traffic volume level in week $i$ is lower than the annual average level. If $\lambda_i > 1$, the provincial traffic volume level in week $i$ is higher than the annual average level. Based on this, we chose $\lambda_i = 1$ as the PEN patency degree threshold.

Then, to calculate the system entropy value, we introduced the patency degree deviation $\Delta\lambda_i$, which is defined as follows:

$$\Delta\lambda_i = max[\lambda_i - 1, \ 0]. \tag{5}$$

The patency degree deviation is 0 when $\lambda_i < 1$. Let $R_i$ be the patency degree deviation sequence of the PEN, $P_i$ be the mapping sequence of $Ri$, and $Pi$ and $Ri$ are respectively defined as follows:

$$R_i = \{\Delta\lambda_i \mid i = 1, \ 2, \ 3, \ldots, \ n\}, \tag{6}$$

$$P_i = \left\{ \Delta\lambda_i / \sum_{i=1}^{n} \Delta\lambda_i \ \middle| \ i = 1, \ 2, \ 3, \ldots, \ n \right\}, \tag{7}$$

where $0 \leq P_i \leq 1$, $\sum_{i=1}^{n} P_i = 1$, $n$ represents the number of weeks in the studied year, and $n = 26$.

Then, for the PEN, the system entropy $S_i(t)$ in week $i$ is as follows:

$$S_i(t) = -C \times K_i \times P_i \times \log_{10} P_i, \tag{8}$$

where $C$ is a constant, and $K_i$ is the system entropy weight in week $i$. As traffic interruption means the failure of traffic function of a road section and affects the traffic operation status of the adjacent roads, it has the greatest impact on the operation orderliness. Therefore, we finally chose WTIT as the key factor to determine the system entropy weight of each week. $K_i$ is allocated proportionally on the basis of WTIT. The calculation formula of $K_i$ is as follows:

$$K_i = \frac{W_i TIT}{\sum_{i=1}^{n} W_i TIT}, \tag{9}$$

where $W_i TIT$ is the WTIT in week $i$, and $\sum_{i=1}^{n} K_i = 1$. On this basis, the annual system entropy $S(t)$ of the PEN in a certain year is calculated as follows:

$$S(t) = -C \sum_{i=1}^{n} (K_i \times P_i \times \log_{10} P_i). \tag{10}$$

Entropy is essentially a relative quantity rather than an absolute quantity, and the function of constant $C$ is mainly to adjust the order of magnitude of system entropy. Hence, to facilitate the display of the entropy calculation results, $C = 100$ was taken in our research.

## 3. Results

The first year of the large-scale COVID-19 outbreak that caught the world by shock was 2020. Based on the traffic data analysis and interpretation, we studied the PEN operation quality in Shaanxi Province from the service level, safety level, and operation orderliness perspective.

### 3.1. Analysis of the PEN Service Level

Figures 2–4 show the boxplots of the PEN service level evaluation parameters (WADT, WTIT, and WTCT). Figures 5–7 depict the dot-line plots of the fixed base growth rate.

Table 3 displays the descriptive statistical results of WADT, WTIT, and WTCT, and Table 4 presents the analysis results of the fixed base growth rate.

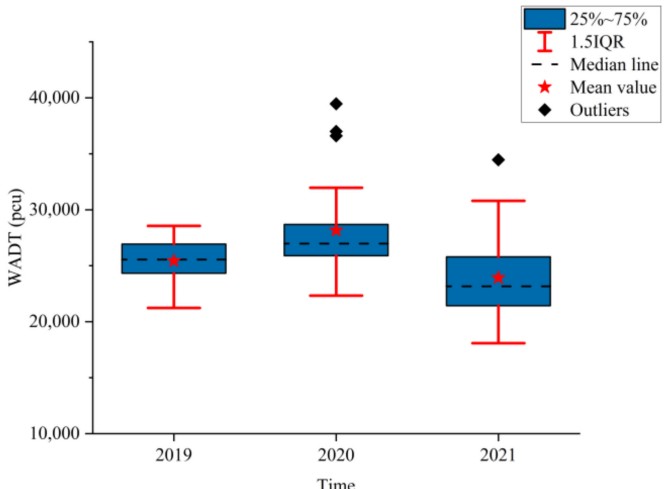

**Figure 2.** WADT.

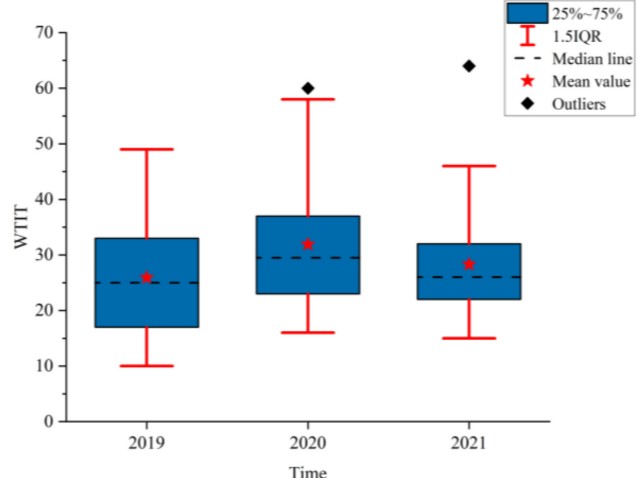

**Figure 3.** WTIT.

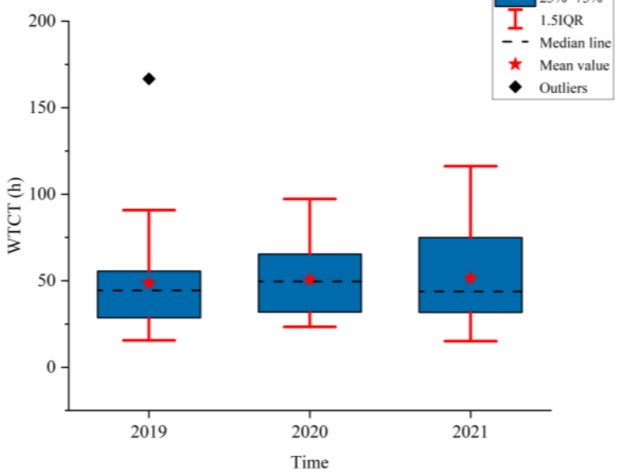

**Figure 4.** WTCT.

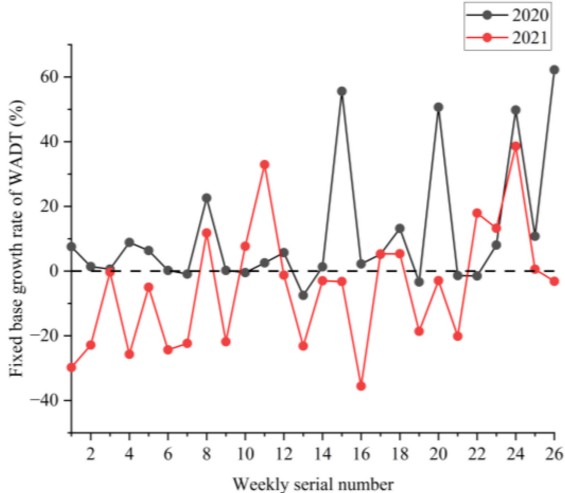

**Figure 5.** Fixed base growth rate of WADT with 2019 as the base year.

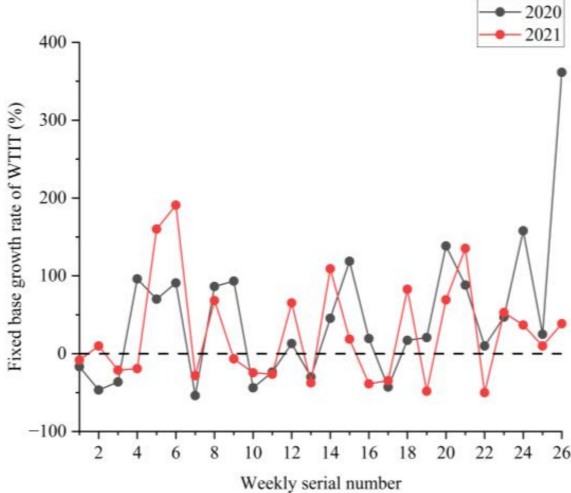

**Figure 6.** Fixed base growth rate of WTIT with 2019 as the base year.

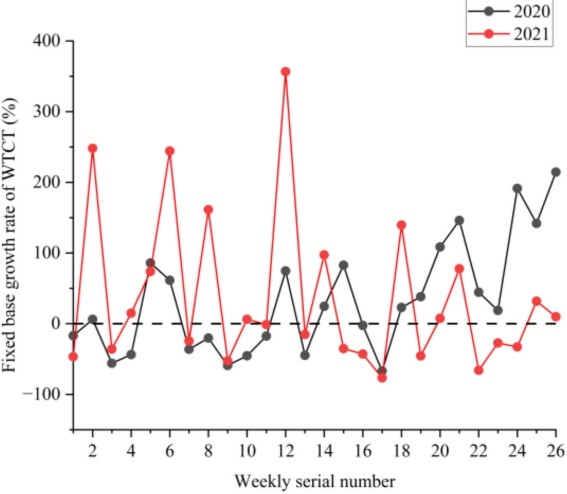

**Figure 7.** Fixed base growth rate of WTCT with 2019 as the base year.

**Table 3.** The descriptive statistical results of the PEN service level evaluation parameters.

| Parameter | Mean Value | | | Standard Deviation | | |
|---|---|---|---|---|---|---|
| | 2019 | 2020 | 2021 | 2019 | 2020 | 2021 |
| WADT (pcu) | 25,432 | 28,167 | 23,923 | 2093 | 4108 | 3919 |
| WTIT | 26 | 32 | 28 | 10.21 | 12.10 | 10.58 |
| WTCT (h) | 48.52 | 50.62 | 51.48 | 30.95 | 20.75 | 29.55 |

**Table 4.** The analysis results of the fixed base growth rate of the PEN service level evaluation parameters (with 2019 as the base year).

| Parameter | Time | Positive Growth Times | Proportion (%) | Negative Growth Times | Proportion (%) | Zero Growth Times | Proportion (%) |
|---|---|---|---|---|---|---|---|
| WADT | 2020 | 20 | 76.92 | 6 | 23.08 | 0 | 0.00 |
| | 2021 | 9 | 34.62 | 17 | 65.38 | 0 | 0.00 |
| WTIT | 2020 | 18 | 69.23 | 8 | 30.77 | 0 | 0.00 |
| | 2021 | 14 | 53.85 | 12 | 46.15 | 0 | 0.00 |
| WTCT | 2020 | 15 | 57.69 | 11 | 42.31 | 0 | 0.00 |
| | 2021 | 13 | 50.00 | 13 | 50.00 | 0 | 0.00 |

The analysis results in Figure 2 show that the box width and the distribution range of 1.5 IQR were the narrowest without outliers. This result indicates that before the COVID-19 outbreak, the distribution of traffic volume was relatively concentrated with low fluctuation. With more outliers, both the box width and 1.5 IQR range of the WADT boxplot in 2020 were larger than those in 2019, and the standard deviation was 4108 pcu, reaching the maximum within the three years. This finding indicates that in the first year of the COVID-19 outbreak, the fluctuation of traffic volume reached the maximum under the impact of pandemic. According to the analysis results, the mean value of WADT in 2020 was higher than that in 2019 and 2021. This is mainly because in the early stage of the outbreak in 2020, passenger and freight transportation at the social level in Shaanxi Province was greatly affected by the pandemic prevention and control policies in a period of time, and it was impossible to operate normally for relative enterprises. As the traffic restriction measures were gradually reduced and lifted, relevant enterprises were eager to resume production due to their interests, and the demand for passenger and freight transportation increased sharply in a short time, resulting in an appreciable increase in the WADT of PEN, which was higher than that in the same period before and after the outbreak. With the operation and production of enterprises gradually returning to normal levels, and subject to the government's regulation, the traffic volume of expressways returned to a relatively stable state and was closer to that of previous years. The analysis results in Table 3 show that the growth rate of WADT standard deviations in 2020 and 2021 reached 96.27% and 87.24%, compared to that in 2019, respectively. This finding indicates that the dispersion and fluctuation of traffic volume distribution increased appreciably in the COVID-19 pandemic context.

The analysis results In Figure 3 and Table 3 show that in 2020, the mean, median, and standard deviation of WTIT reached 32, 29.5, and 12.10, respectively, all reaching the maximum within the three years. In addition, the 1.5IQR distribution range of the WTIT boxplot increased compared to that in 2019. This finding indicates that the frequency of traffic interruption accidents increased after the COVID-19 outbreak, and the fluctuation of traffic interruption reached the maximum within the three years.

In 2021, the mean, median, and standard deviation of WTIT was 28, 26, and 10.58, with a decrease of 12.50%, 11.86%, and 12.56%, compared to that in 2020, respectively. However, they were still higher than the normal level in 2019 before the COVID-19 outbreak. Furthermore, the 1.5IQR distribution range of the WTIT boxplot in 2021 decreased compared to that in 2020. This finding indicates that the frequency of traffic interruption accidents and its fluctuation decreased with the gradual control of the pandemic.

The analysis results in Figure 4 and Table 3 show that from 2019 to 2021, the WTCT mean values were 48.52, 50.62, and 51.48 h, respectively, with an increasing trend year by year. Meanwhile, the box width of the WTCT boxplot kept increasing within the three years. This result indicates that the traffic control time and its fluctuation kept increasing in the COVID-19 pandemic context.

The analysis results in Table 4 show that, in 2020, the proportion of positive growth in the fixed base growth rate of WADT, WTIT, and WTCT was 76.92%, 69.23%, and 57.69%, respectively, all of which were higher than in 2021. This finding indicates that in the first year of the COVID-19 outbreak, the traffic volume, traffic interruptions, and traffic control time showed an increasing trend on the whole. In 2021, the proportion of positive growth of WADT, WTIT, and WTCT decreased by 42.3%, 15.38%, and 7.69%, compared to that in 2020, respectively. This result indicates that the impact of the pandemic on the PEN service level parameters began to decrease with the gradual control of COVID-19, and the growth trend of the PEN service level parameters slowed down.

The analysis of the aforementioned statistical parameters shows that the distribution of the PEN traffic volume was relatively stable, and the overall fluctuation was small in the year before the COVID-19 outbreak. However, the COVID-19 outbreak impacted the traffic volume level and caused a large fluctuation of it to some extent. Moreover, the pandemic increased traffic interruption times and their fluctuation to a certain extent, and prolonged traffic control time. The PEN service level made limited responses to the impact of COVID-19. The increase in traffic volume fluctuation, traffic interruption times, and traffic control time led to a decrease in the PEN service level compared with the year before the COVID-19 outbreak. As the pandemic was gradually brought under control, traffic interruption times and traffic volume fluctuation of the PEN decreased, and the service level recovered to some extent.

### 3.2. Analysis of the PEN Safety Level

Figures 8–10 show the boxplots of the PEN safety level evaluation parameters (WTA, WTI, and WTD). Figures 11–13 depict the dot-line plots of the fixed base growth rate, with 2019 as the base year. Table 5 shows the descriptive statistical results of WTA, WTI, and WTD, and Table 6 shows the analysis results of the fixed base growth rate. Then, Table 7 shows the calculation results of the PEN accident rate and casualty rate, and Table 8 shows the descriptive statistical results of the PEN accident rate and casualty rate.

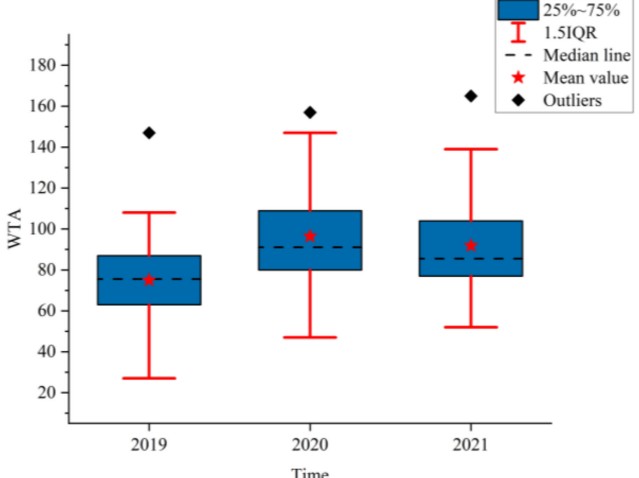

**Figure 8.** WTA.

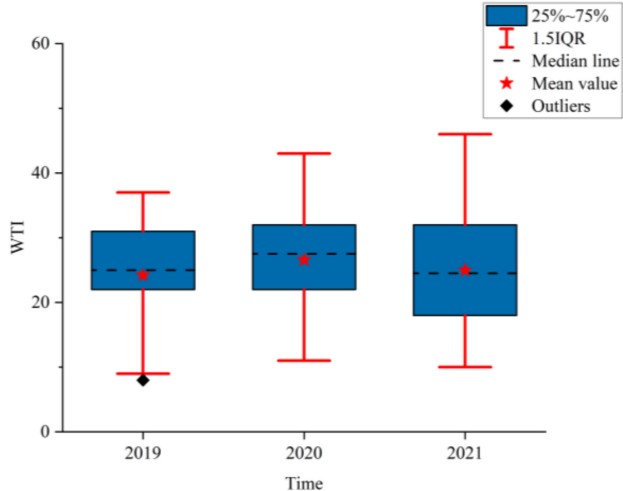

**Figure 9.** WTI.

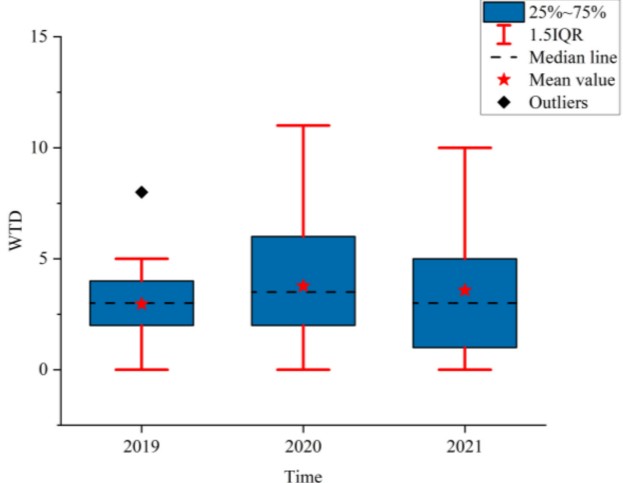

**Figure 10.** WTD.

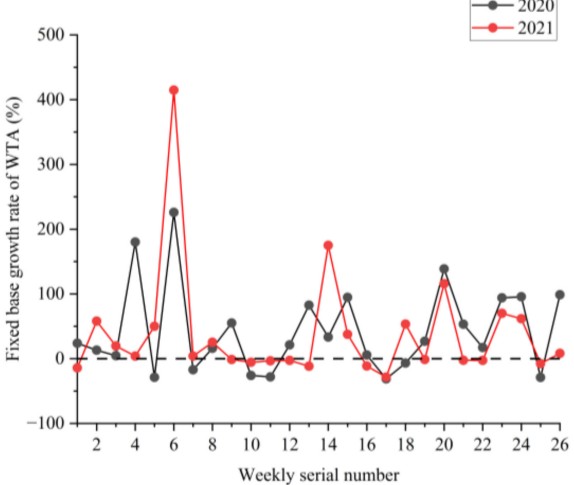

**Figure 11.** Fixed base growth rate of WTA with 2019 as the base year.

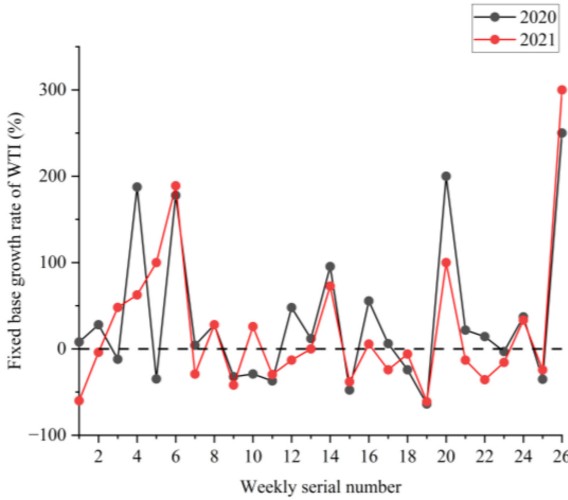

**Figure 12.** Fixed base growth rate of WTI with 2019 as the base year.

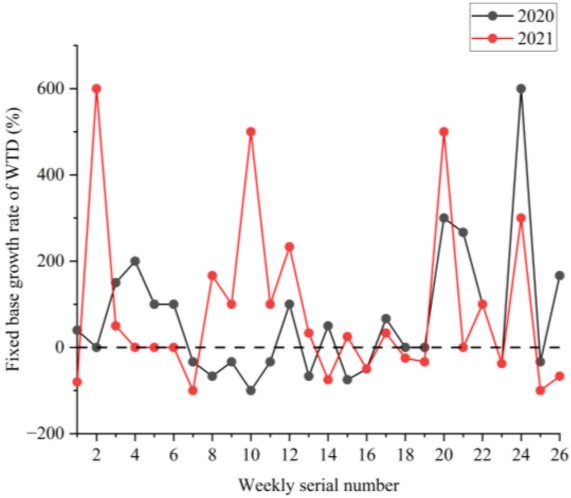

**Figure 13.** Fixed base growth rate of WTD with 2019 as the base year.

**Table 5.** The descriptive statistical results of the PEN safety level evaluation parameters.

| Parameter | Mean Value | | | Standard Deviation | | |
|---|---|---|---|---|---|---|
| | **2019** | **2020** | **2021** | **2019** | **2020** | **2021** |
| WTA | 75 | 96.42 | 91.85 | 22.67 | 26.98 | 25.3 |
| WTI | 24.23 | 26.58 | 24.96 | 8.08 | 7.52 | 9.27 |
| WTD | 2.96 | 3.77 | 3.58 | 1.61 | 2.63 | 2.56 |

**Table 6.** The analysis results of the fixed base growth rate of the PEN safety level evaluation parameters (with 2019 as the base year).

| Parameter | Time | Positive Growth Times | Proportion (%) | Negative Growth Times | Proportion (%) | Zero Growth Times | Proportion (%) |
|---|---|---|---|---|---|---|---|
| WTA | 2020 | 19 | 73.08 | 7 | 26.92 | 0 | 0.00 |
| | 2021 | 14 | 53.85 | 12 | 46.15 | 0 | 0.00 |
| WTI | 2020 | 16 | 61.54 | 10 | 38.46 | 0 | 0.00 |
| | 2021 | 11 | 42.31 | 14 | 53.85 | 1 | 3.85 |
| WTD | 2020 | 13 | 50.00 | 10 | 38.46 | 3 | 11.54 |
| | 2021 | 12 | 46.15 | 9 | 34.62 | 5 | 19.23 |

**Table 7.** The calculation results of the PEN accident rate and casualty rate.

| Weekly Serial Number | Weekly Traffic Accident Rate (1/10,000) | | | Weekly Traffic Injury Rate (1/10,000) | | | Weekly Traffic Death Rate (1/10,000) | | |
|---|---|---|---|---|---|---|---|---|---|
| | **2019** | **2020** | **2021** | **2019** | **2020** | **2021** | **2019** | **2020** | **2021** |
| 1 | 9.89 | 6.46 | 6.71 | 2.22 | 2.24 | 1.60 | 0.20 | 0.32 | 0.26 |
| 2 | 3.87 | 5.09 | 10.96 | 1.42 | 2.73 | 2.52 | 0.26 | 0.38 | 0.07 |
| 3 | 3.40 | 4.25 | 4.83 | 1.28 | 0.43 | 0.82 | 0.24 | 0.04 | 0.31 |
| 4 | 2.56 | 4.05 | 5.69 | 0.64 | 1.27 | 1.32 | 0.06 | 0.15 | 0.36 |
| 5 | 4.34 | 5.32 | 4.85 | 0.47 | 1.01 | 1.94 | 0.18 | 0.29 | 0.06 |
| 6 | 4.55 | 5.94 | 5.31 | 1.81 | 1.65 | 1.74 | 0.00 | 0.27 | 0.14 |
| 7 | 5.36 | 3.43 | 4.92 | 2.13 | 1.25 | 1.60 | 0.17 | 0.10 | 0.00 |
| 8 | 5.27 | 4.34 | 7.68 | 1.98 | 1.32 | 1.76 | 0.24 | 0.21 | 0.17 |
| 9 | 3.75 | 2.51 | 5.92 | 1.31 | 0.80 | 2.75 | 0.06 | 0.11 | 0.06 |
| 10 | 4.88 | 5.08 | 5.86 | 1.40 | 1.23 | 2.08 | 0.11 | 0.28 | 0.17 |
| 11 | 6.02 | 4.48 | 5.28 | 1.73 | 1.23 | 2.02 | 0.28 | 0.00 | 0.26 |
| 12 | 4.16 | 4.94 | 3.43 | 1.53 | 1.78 | 0.84 | 0.11 | 0.22 | 0.19 |
| 13 | 4.90 | 3.44 | 3.57 | 1.49 | 0.91 | 0.79 | 0.17 | 0.11 | 0.25 |
| 14 | 2.89 | 5.19 | 4.34 | 1.85 | 1.66 | 1.38 | 0.46 | 0.27 | 0.26 |
| 15 | 4.64 | 4.39 | 5.20 | 1.55 | 1.62 | 1.77 | 0.19 | 0.05 | 0.44 |
| 16 | 3.86 | 4.43 | 3.81 | 1.18 | 1.66 | 1.04 | 0.15 | 0.29 | 0.52 |
| 17 | 1.35 | 4.39 | 9.19 | 0.45 | 1.25 | 1.72 | 0.05 | 0.10 | 0.07 |
| 18 | 3.84 | 3.21 | 5.13 | 1.21 | 1.27 | 1.10 | 0.15 | 0.10 | 0.00 |
| 19 | 3.18 | 3.66 | 3.88 | 1.26 | 1.27 | 0.72 | 0.25 | 0.33 | 0.07 |
| 20 | 4.48 | 4.63 | 6.16 | 0.92 | 1.39 | 1.50 | 0.20 | 0.10 | 0.16 |
| 21 | 3.55 | 3.96 | 7.27 | 1.29 | 1.62 | 1.60 | 0.05 | 0.05 | 0.47 |
| 22 | 4.14 | 5.43 | 5.02 | 1.91 | 0.71 | 0.91 | 0.16 | 0.16 | 0.13 |
| 23 | 4.19 | 6.51 | 5.11 | 1.25 | 1.54 | 1.36 | 0.16 | 0.61 | 0.20 |
| 24 | 4.66 | 9.19 | 5.36 | 1.35 | 1.64 | 1.76 | 0.16 | 0.06 | 0.28 |
| 25 | 4.36 | 6.74 | 5.51 | 1.69 | 1.14 | 1.25 | 0.16 | 0.11 | 0.42 |
| 26 | 2.88 | 7.42 | 4.04 | 0.46 | 1.22 | 1.01 | 0.12 | 0.32 | 0.16 |
| Total for the year | 4.21 | 4.89 | 5.48 | 1.36 | 1.35 | 1.49 | 0.17 | 0.19 | 0.21 |

**Table 8.** The descriptive statistical results of the PEN accident rate and casualty rate.

| Index | Mean Value (1/10,000) | | | Standard Deviation (1/10,000) | | |
|---|---|---|---|---|---|---|
| | **2019** | **2020** | **2021** | **2019** | **2020** | **2021** |
| Weekly traffic accident rate | 4.27 | 4.94 | 5.58 | 1.50 | 1.45 | 1.70 |
| Weekly traffic injury rate | 1.38 | 1.38 | 1.50 | 0.49 | 0.47 | 0.52 |
| Weekly traffic death rate | 0.17 | 0.19 | 0.21 | 0.09 | 0.14 | 0.14 |

The analysis results in Figure 8 and Table 5 show that in 2020, the mean, median, and standard deviation of WTA was 96.42, 91, and 26.98, respectively, reaching the maximum within the three years. Meanwhile, their growth rate was 28.56%, 20.53%, and 19.01%, compared to that in 2019, respectively. The box width and 1.5IQR distribution range of the WTA boxplot were the largest in 2020. This finding indicates that in the first year of the COVID-19 outbreak, the number of traffic accidents, including their fluctuation, reached the maximum, and their distributions were relatively discrete.

In 2021, the mean, median, and standard deviation of WTA was 91.85, 85.5, and 25.3, with a decrease of 4.74%, 6.04%, and 6.23%, compared to that in 2020, respectively. However, they still maintained a growth rate of 22.47%, 13.25%, and 11.60%, compared to that in 2019, respectively. The box width of the WTA boxplot decreased, whereas it was still higher than that in 2019. This result indicates that the number of traffic accidents began to decrease in 2021 with the gradual control of the pandemic. Its distribution was more concentrated with a reduction of the overall fluctuation. However, the number of traffic accidents did not return to the pre-pandemic level.

The analysis results in Figure 9 and Table 5 show that in 2020, the mean and median of WTI was 26.58 and 27.50, respectively, reaching the maximum within the three years. Furthermore, their growth rate reached 9.70% and 10.00%, compared to that in 2019, respectively. The standard deviation of WTI was 7.52 in 2020, the minimum within the three years, with a decrease of 6.93% compared to that in 2019. This finding indicates that in the first year of the COVID-19 outbreak, the number of traffic injuries increased, with a decrease in fluctuation and an increase in distribution concentration.

The analysis results in Figure 10 and Table 5 show that in 2020, the mean, median, and standard deviation of WTD was 3.77, 4, and 2.63, respectively, all reaching the maximum within the three years. Moreover, the growth rate was 27.36%, 33.33%, and 63.35%, compared to that in 2019, respectively. This result indicates that the number of traffic deaths, including the fluctuation, increased appreciably in the COVID-19 pandemic context.

The analysis results in Table 6 show that the growth in the fixed base growth rate of WTA, WTI, and WTD in 2020 was 73.58%, 61.54%, and 50.00%, respectively, all of which were higher than those in 2021. Furthermore, in 2021, the proportion of positive growth in the fixed base growth rate of the aforementioned parameters decreased by 19.73%, 19.23%, and 3.85%, respectively. This result indicates that after the COVID-19 outbreak, the number of traffic accidents, traffic injuries, and traffic deaths increased simultaneously from 2020 to 2021, compared to those in 2019. With the pandemic gradually controlled, the growth trend of the PEN safety level parameters slowed down, and the impact of the pandemic on the PEN safety level began to decrease.

The analysis results in Tables 7 and 8 show that compared with 2019, the annual traffic accident rate in 2020 and 2021 increased by 16.15% and 30.17%, respectively. Besides, the average traffic accident rate of each week increased by 15.69% and 30.68%, respectively. This finding indicates that after the COVID-19 outbreak, the traffic accident rate of the PEN continued to increase within two of the years, and the PEN safety level continued to decrease. Compared to 2019, The annual traffic injury rate in 2020 and 2021 increased by −0.74% and 9.56%, and the annual traffic death rate increased by 11.76% and 23.53%, respectively. Furthermore, compared with 2019, the average weekly traffic injury rate increased by 0% and 8.70%, and the average weekly traffic death rate increased by 11.76% and 23.53%, respectively. This result indicates that after the COVID-19 outbreak, the traffic accident death rate increased appreciably, and the increase in traffic death rate was appreciably higher than that of traffic injury rate. Therefore, the traffic accident severity was exacerbated appreciably in the COVID-19 context.

Compared with 2019, the standard deviation of traffic accident rate and traffic injury rate in each week in 2020 decreased by 3.33% and 4.08%, whereas the standard deviation of traffic death rate in each week increased by 55.56%, respectively. This result indicates that the fluctuation of the traffic accident rate and traffic injury rate did not change much after the COVID-19 outbreak. However, the fluctuation of traffic death rate increased appreciably.

The analysis of the aforementioned statistical parameters shows that the COVID-19 outbreak impacted the PEN safety level, increased the fluctuation of traffic accidents, the number of traffic accidents, accident injuries, and accident deaths. The outbreak appreciably increased the death rate of traffic accidents and exacerbated the traffic accident severity. In summary, The PEN safety level made limited responses to the impact of the COVID-19 pandemic. As the pandemic was gradually brought under control, the number of accidents, injuries, and deaths decreased, whereas the accident rate and accident casualty rate remained high.

### 3.3. Analysis of the PEN Operation Orderliness

Table 9 shows the calculation results of the weekly system entropy weight of the PEN from 2019 to 2021. Table 10 shows the calculation results of the patency degree, patency degree deviation, and system entropy in each week of the PEN from 2019 to 2021. Then, Table 11 shows the descriptive statistical results of system entropy.

**Table 9.** The calculation results of the weekly system entropy weight of the PEN.

| Weekly Serial Number | $K_i$ | | | Weekly Serial Number | $K_i$ | | |
|---|---|---|---|---|---|---|---|
| | 2019 | 2020 | 2021 | | 2019 | 2020 | 2021 |
| 1 | 0.04 | 0.03 | 0.03 | 14 | 0.01 | 0.02 | 0.02 |
| 2 | 0.02 | 0.03 | 0.04 | 15 | 0.02 | 0.04 | 0.03 |
| 3 | 0.01 | 0.03 | 0.02 | 16 | 0.02 | 0.02 | 0.03 |
| 4 | 0.01 | 0.03 | 0.02 | 17 | 0.01 | 0.02 | 0.03 |
| 5 | 0.01 | 0.05 | 0.02 | 18 | 0.03 | 0.02 | 0.03 |
| 6 | 0.02 | 0.04 | 0.02 | 19 | 0.02 | 0.02 | 0.02 |
| 7 | 0.02 | 0.02 | 0.02 | 20 | 0.02 | 0.03 | 0.02 |
| 8 | 0.03 | 0.04 | 0.06 | 21 | 0.02 | 0.01 | 0.03 |
| 9 | 0.01 | 0.02 | 0.02 | 22 | 0.02 | 0.03 | 0.01 |
| 10 | 0.03 | 0.02 | 0.02 | 23 | 0.01 | 0.03 | 0.04 |
| 11 | 0.03 | 0.02 | 0.03 | 24 | 0.03 | 0.03 | 0.02 |
| 12 | 0.02 | 0.03 | 0.01 | 25 | 0.02 | 0.05 | 0.03 |
| 13 | 0.03 | 0.02 | 0.02 | 26 | 0.02 | 0.05 | 0.02 |

**Table 10.** The calculation results of the patency degree, patency degree deviation, and system entropy in each week of the PEN from 2019 to 2021.

| Weekly Serial Number | $\lambda_i$ | | | $\Delta\lambda_i$ | | | $S_i(t)$ | | |
|---|---|---|---|---|---|---|---|---|---|
| | 2019 | 2020 | 2021 | 2019 | 2020 | 2021 | 2019 | 2020 | 2021 |
| 1 | 0.83 | 0.79 | 0.93 | 0.00 | 0.00 | 0.00 | 0.00 | 0.00 | 0.00 |
| 2 | 0.87 | 0.80 | 0.90 | 0.00 | 0.00 | 0.00 | 0.00 | 0.00 | 0.00 |
| 3 | 0.93 | 1.30 | 0.95 | 0.00 | 0.30 | 0.00 | 0.00 | 0.46 | 0.00 |
| 4 | 0.97 | 1.31 | 1.00 | 0.00 | 0.31 | 0.00 | 0.00 | 0.42 | 0.00 |
| 5 | 0.96 | 1.40 | 0.98 | 0.00 | 0.40 | 0.00 | 0.00 | 0.86 | 0.00 |
| 6 | 0.84 | 1.13 | 1.24 | 0.00 | 0.13 | 0.24 | 0.00 | 0.45 | 0.27 |
| 7 | 0.98 | 0.98 | 1.04 | 0.00 | 0.00 | 0.04 | 0.00 | 0.00 | 0.08 |
| 8 | 0.94 | 0.96 | 1.05 | 0.00 | 0.00 | 0.05 | 0.00 | 0.00 | 0.25 |
| 9 | 0.99 | 0.95 | 1.00 | 0.00 | 0.00 | 0.00 | 0.00 | 0.00 | 0.00 |
| 10 | 1.00 | 0.91 | 1.06 | 0.00 | 0.00 | 0.06 | 0.01 | 0.00 | 0.12 |
| 11 | 1.01 | 0.91 | 1.15 | 0.01 | 0.00 | 0.15 | 0.05 | 0.00 | 0.26 |
| 12 | 1.03 | 0.91 | 1.29 | 0.03 | 0.00 | 0.29 | 0.09 | 0.00 | 0.17 |
| 13 | 1.02 | 0.94 | 1.44 | 0.02 | 0.00 | 0.44 | 0.09 | 0.00 | 0.34 |
| 14 | 0.97 | 0.95 | 1.17 | 0.00 | 0.00 | 0.17 | 0.00 | 0.00 | 0.23 |
| 15 | 0.91 | 1.00 | 1.08 | 0.00 | 0.00 | 0.08 | 0.00 | 0.03 | 0.20 |
| 16 | 1.09 | 1.04 | 1.15 | 0.09 | 0.04 | 0.15 | 0.17 | 0.11 | 0.30 |
| 17 | 1.12 | 1.02 | 0.90 | 0.12 | 0.02 | 0.00 | 0.09 | 0.04 | 0.00 |
| 18 | 1.11 | 1.00 | 0.92 | 0.11 | 0.00 | 0.00 | 0.32 | 0.00 | 0.00 |
| 19 | 1.11 | 1.08 | 0.83 | 0.11 | 0.08 | 0.00 | 0.20 | 0.14 | 0.00 |
| 20 | 1.10 | 1.02 | 0.76 | 0.10 | 0.02 | 0.00 | 0.24 | 0.09 | 0.00 |
| 21 | 1.09 | 1.00 | 0.90 | 0.09 | 0.00 | 0.00 | 0.22 | 0.00 | 0.00 |
| 22 | 1.06 | 0.92 | 0.92 | 0.06 | 0.00 | 0.00 | 0.16 | 0.00 | 0.00 |
| 23 | 1.03 | 0.92 | 0.88 | 0.03 | 0.00 | 0.00 | 0.06 | 0.00 | 0.00 |
| 24 | 1.04 | 0.87 | 0.85 | 0.04 | 0.00 | 0.00 | 0.16 | 0.00 | 0.00 |
| 25 | 1.03 | 0.93 | 0.86 | 0.03 | 0.00 | 0.00 | 0.11 | 0.00 | 0.00 |
| 26 | 0.97 | 0.96 | 0.77 | 0.00 | 0.00 | 0.00 | 0.00 | 0.00 | 0.00 |
| $\Sigma$ | | | | | | | 1.96 | 2.60 | 2.23 |

**Table 11.** The descriptive statistical results of system entropy.

| Time | Minimum Value | Maximum Value | Mean Value | Standard Deviation |
|---|---|---|---|---|
| 2019 | 0.00 | 0.32 | 0.08 | 0.09 |
| 2020 | 0.00 | 0.86 | 0.10 | 0.21 |
| 2021 | 0.00 | 0.34 | 0.09 | 0.12 |

The analysis results in Tables 10 and 11 show that the total entropy of the PEN in 2020 was 2.60, reaching the maximum within the three years, with a growth rate of 32% compared to that in 2019. Meanwhile, the mean and standard deviation of entropy of each week in 2020 increased by 25% and 133% compared to those in 2019, respectively. This finding indicates that in the first year of the COVID-19 outbreak, the total entropy of the PEN increased appreciably, the fluctuation sharply increased, and the operation orderliness decreased appreciably. The total entropy of the PEN in 2021 decreased by 14% compared to that in 2020 but still maintained a growth rate of 13% compared to that in 2019. This result indicates that with the control of COVID-19, the PEN operation orderliness improved but did not yet recover to the level of 2019, before the pandemic.

In 2021, the average weekly entropy of the PEN decreased by 10% compared to that in 2020, whereas with a growth rate of 13% compared to that in 2019. The standard deviation of the weekly entropy in 2021 decreased by 43% compared to that in 2021, whereas with a growth rate of 33% compared to that in 2019. This finding indicates that, with the gradual control of the COVID-19 pandemic, the PEN operation orderliness improved, its fluctuation diminished, and the overall operation stability of the PEN improved. However, a gap still exists from the normal level in 2019.

Based on the above analysis, we have reason to believe that after the COVID-19 outbreak, the PEN operation orderliness had a strong response to the impact of COVID-19. Compared with the previous year, the operation orderliness decreased appreciably, and its fluctuation increased appreciably. Hence, the overall stability of the PEN was poorer. With the gradual control of the pandemic, the operation orderliness improved in 2021 but did not yet recover to the normal level before the pandemic.

## 4. Discussion

In this study, the limited responses of the PEN operation quality to the impact of the COVID-19 pandemic in Shaanxi Province were examined. Using descriptive statistics and dynamic analysis, we discussed the PEN service and safety levels in the COVID-19 context. By dissipative structure theory and entropy theory, based on the operation orderliness perspective, an evaluation model was established to analyze the PEN operation orderliness in the COVID-19 context. Through the above analysis, we confirmed the adverse impacts of COVID-19 on the operation quality of the PEN.

The results in our study show that the operation quality of the PEN in Shaanxi Province made limited responses to the impact of COVID-19, and the pandemic caused a reduction in the PEN operation quality to a certain extent, including the reduction in service level, safety level, and operation orderliness. Among them, the decrease in service level was mainly manifested by the increase in traffic volume fluctuation, traffic interruption times, and traffic control time. The decrease in safety level was mainly reflected in the increase in traffic accidents, traffic injuries, traffic deaths, the appreciable increase in traffic accident rate, traffic accident death rate, and the aggravation of traffic accident severity. The decrease in operation orderliness was mainly manifested by the increase in system entropy, the decrease in orderliness degree, and the intensification of the fluctuation. With the gradual control of COVID-19, the service level and operation orderliness began to recover, but the safety level did not improve. The recovery of the service level was mainly manifested in the reduction in traffic interruption times and the traffic volume fluctuation. The recovery of the operation orderliness was mainly manifested in the reduction in system entropy and its fluctuation. In terms of the safety level, the number of traffic accidents, traffic injuries, and traffic deaths decreased, but the traffic accident rate and accident casualty rate remained high.

This study only examined the limited responses of the PEN operation quality to the impact of COVID-19 and could not further explore the deep-seated reasons for the limited responses because of the limited data. However, based on the prevention and control measures for COVID-19 in Shaanxi Province, referring to the research results of other

researchers, we could still make some reasonable analyses. These analyses could be the reasons for the changes of the PEN operation quality to a certain extent.

First, in the early stage of the pandemic, the Shaanxi provincial government adopted strict traffic restriction measures, and passenger and freight transportation on the PEN could not be carried out normally. As the pandemic gradually came under control, the traffic restriction measures were gradually reduced and lifted, resulting in a sharp increase in the demand for passenger and freight transportation on the PEN. Then, the traffic volume increased significantly in a short time and was higher than that in the same period before the pandemic. This event led to a decline in the PEN operation orderliness to a certain extent, increased the frequency of traffic accidents, and then increased traffic interruptions and traffic control time. As the demand for passenger and freight transportation was alleviated, the traffic volume of the PEN gradually reached a level close to that in the previous year, and the fluctuation reduced.

Second, the pandemic and its prevention and control measures brought non-negligible impacts on drivers. For example, long-term home quarantine led to drivers' psychological depression and then increased dangerous driving behaviors, such as speeding and aggressive driving, resulting in an increase in traffic accident rates, traffic injuries, traffic accident death rates and severity. These may be the deep-seated reasons for the decline of the PEN operation quality in the COVID-19 context.

In summary, the impacts of the COVID-19 pandemic on the expressway traffic were objective facts, but we did not learn much about the responses of the PEN operation quality to COVID-19, and how the PEN operation quality evolved at the macro level in the pandemic context. These questions needed to be explored through data analysis, which was exactly what this study attempted to address. For the traffic management departments, it is necessary to understand the impacts of the pandemic on traffic quality. The traffic departments could take effective measures to ensure the stable, safe and orderly operation of the expressway network only by fully grasping the response characteristics of the PEN operation quality to the impact of COVID-19.

*Limitations*

First, limited by the availability of traffic data, to improve the reliability of the research results, we excluded the PEN operation data for some weeks from 2019 to 2021. We retained the data with overlapping time points within the three years and finally obtained the traffic data of 26 weeks of each year as the research material. Although the sample size was reduced, the weekly sequences of these data were completely corresponding, which could ensure the typicality, comparability, and effectiveness of the research materials. Thus, the research results could reflect the PEN operation quality in Shaanxi Province more truly and effectively. Undoubtedly, the results of this study would be more comprehensive and accurate if the complete PEN operation data for 52 weeks in the whole year could be used.

Second, the changes of the PEN operation parameters are random to a certain extent, and our research could not completely rule out the interference of randomness. However, quantitative analysis and comprehensive comparison of multiple parameters and indicators of the PEN operation could reduce the randomness interference to a certain extent by incorporating the data of three consecutive years since the COVID-19 outbreak into the research.

Finally, there were other factors besides COVID-19 that would affect the PEN operation quality, but this study did not further analyze them. These limitations need to be further considered in future studies to reflect the PEN operation quality more accurately in the COVID-19 context.

## 5. Conclusions

Learning the responses of the PEN operation quality to the impact of COVID-19 is very important. It can help traffic management departments to take reasonable and effective measures to ensure the PEN service level, safety level, and operation orderliness in the

COVID-19 context and finally promote the improvement of the PEN operation quality. For example, during the pandemic prevention and control period, traffic management departments should strengthen the control of vehicle operation, such as strictly controlling speeding behavior by implementing stricter speed restriction polices. Different speed limit values should be adopted according to the changes of traffic volume to ensure the maximum safety and stability of traffic flow on the PEN. These are some effective measures that traffic management departments can take.

On the one hand, we hope to win the battle against the COVID-19 pandemic as soon as possible. On the other hand, we also need to use more diversified and comprehensive traffic data to deeply explore the impact mechanism of major public health emergencies, such as the COVID-19 pandemic, on traffic system operations in future research. Thus, we can provide more reliable theoretical bases and decision supports for traffic management departments in formulating reasonable and effective traffic management measures to promote the improvement of traffic operation quality.

**Author Contributions:** Conceptualization, Y.M. and J.X.; methodology, Y.M.; software, Y.M.; validation, Y.M.; formal analysis, Y.M.; investigation, Y.M.; resources, J.X. and X.T.; data curation, Y.M., C.G. (Chao Gao) and C.G. (Chenwei Gu); writing—original draft preparation, Y.M.; writing—review and editing, Y.M. and J.X.; visualization, Y.M.; supervision, J.X.; project administration, X.T.; funding acquisition, J.X. All authors have read and agreed to the published version of the manuscript.

**Funding:** This research was funded by Scientific Research Project of Department of Transport of Shaanxi Province, grant number 21-03R and The APC was funded by Department of Transport of Shaanxi Province.

**Institutional Review Board Statement:** Not applicable.

**Informed Consent Statement:** Not applicable.

**Data Availability Statement:** Publicly available data sets were analyzed in this study. These data can be found here: [https://jtyst.shaanxi.gov.cn/news/570526375327/3.htm](https://jtyst.shaanxi.gov.cn/news/570526375327/3.htm) (accessed on 10 July 2022)].

**Acknowledgments:** The authors would like to thank the members of China Communications Construction First Highway Consultants Co. Ltd. for their cooperation in the data collection.

**Conflicts of Interest:** The authors declare no conflict of interest. The funders had no role in the design of the study; in the collection, analyses, or interpretation of data; in the writing of the manuscript; or in the decision to publish the results.

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
