# Peer review of "The Limited Responses of Provincial Expressway Network Operation Quality to the Impact of COVID-19: Taking Shaanxi Province as an Example"

_sustainability, doi:10.3390/su141710859_

Round 1

Reviewer 1 Report

The paper is well organised, the topic is interesting, although I have problems with the original dataset. How is it possible that the WADT (pcu) is not going back in 2020 although there were closure due to pandemic. Please control these numbers, or give clear explanation to this phenomena!

The topics related to COVID19 are very fashionable, so please use a wider range of cited papers.

Author Response

Dear reviewer,

Yours sincerely

Reviewer 2 Report

The authors analyze the traffic operation quality of the provincial expressway network in Shaanxi Province in the COVID-19 pandemic context. Their research is based on the network traffic data (weekly average daily traffic, traffic interruption times, control time, accidents, injuries, and deaths), collected from 127 observation sites over three consecutive years. They have systematized the data for the 26 weeks of each year with overlapping time points. Through statistical analysis and modeling (according to the dissipative structure and entropy theory), they have quantified the influence of the pandemic on the network's service level, safety level, and operation orderliness.

The manuscript is clear and presented in a well-structured manner. In the introduction, the authors give a short but sufficient overview of previous research on the main topics and use current references. The research is scientifically sound and the authors have proven the appropriateness of their experimental design. The data is interpreted appropriately and consistently throughout the manuscript, and the results are reproducible based on the details given in the methods section. Discussion and conclusions are consistent with the evidence and arguments presented. Overall, this is very fine work. I do have some specific comments for the authors which I believe will contribute to the manuscript:

Lines 52-54: Please give a summary of the general conclusions of the studies you are referencing.

Lines 65-66: Please state Macioszek's findings.

Line 100: Please add the information on the total length of analysed expressway network.

Line 102: The resolution of Figure 1 is quite poor, please try to enhance it and use a smaller font size for the legend.

Line 158: Please delete the last row of Table 1.

Lines 257-270: It seems to me that this paragraph would fit better after Tables 1 and 2.

Line 227, 344: Please indicate the base year here and in Figures 5, 6, 7, 11, 12, and 13 and Table 4 and 6 captions.

Author Response

Dear reviewer,

Yours Sincerely
